# SM: Bridging the Robustness Gap in Clinical Time Series Analysis via Hierarchical Stability Optimization

## Abstract

Deep learning models for medical time series analysis exhibit a critical reliability gap: high accuracy on curated data does not translate to robustness against real-world noise and device variability. We argue this gap stems from inadequate modeling of hierarchical physiology and training paradigms that neglect clinical stability. We introduce **SM** (**S**tability **M**edical time series classifier), a framework that bridges this gap by synergistically co-designing a novel, physiologically-inspired architecture with a multifaceted stability optimization strategy. Our **S**tability-aware **H**ierarchical **S**patial **M**odulation (SHSM) module mimics clinical reasoning by selectively attending to biomarkers while preserving global waveform morphology. Complementing this, our training objective enforces robust accuracy, output consistency, and knowledge preservation without sacrificing clean-data performance. Extensive evaluations on four medical time series datasets against 11 baselines demonstrate that SM achieves state-of-the-art performance while significantly improving robustness. By unifying architecture and training around the principle of stability, SM provides a systematic framework for building clinically reliable medical AI.

## 1 Introduction

Medical time-series analysis is fundamental to modern clinical diagnosis, involving the examination of sequential health-related data points recorded over time to monitor physiological signals, with modalities like electroencephalography (EEG) and electrocardiography (ECG) offering essential insights into neurological and cardiovascular conditions Badr et al. (2024); Altaheri et al. (2023). This approach is vital for transforming healthcare management by improving patient outcomes, reducing costs, and increasing operational efficiency Liu et al. (2021); Murat et al. (2020). Its applications are extensive, ranging from epidemiology, where it is used to predict disease outbreaks, to hospital administration for forecasting emergency department visits and optimizing resource allocation Li et al. (2025). In direct patient care, advanced machine learning techniques enable the anticipation of critical events such as organ failure or adverse treatment responses, facilitating earlier, life-saving interventions.

However, deploying deep learning models in real-world clinical settings presents significant challenges. Physiological signals acquired in practice exhibit complex noise patterns that systematically deviate from those in laboratory-curated datasets Tzimourta et al. (2021); Al-Zaiti et al. (2023). EEG recordings, for instance, are susceptible to motion artifacts and inter-electrode impedance variations Sanei & Chambers (2013), while ECG measurements suffer from inter-device variability and patient-specific baseline drifts Kiyasseh et al. (2021). Consequently, a critical robustness gap emerges: state-of-the-art models, such as Medformer Wang et al. (2024b), achieve high accuracy on benchmark datasets but exhibit fragile decision boundaries when confronted with realistic perturbations Liu et al. (2021); Murat et al. (2020).

This fragility arises from two fundamental limitations. First, existing architectures inadequately capture the hierarchical organization of physiological patterns that span multiple temporal resolutions Nie et al. (2023); Zhang & Yan (2022). Clinicians routinely integrate information from low-frequency EEG rhythms (e.g., delta waves at 0.5–4 Hz) and high-frequency ECG features (e.g., QRS complexes)

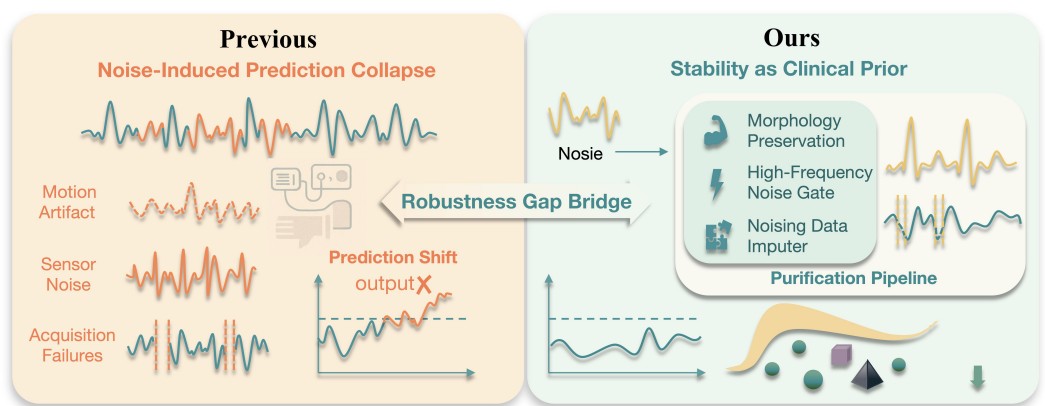

Figure 1: Bridging the robustness gap in medical time series analysis. **Left**: Conventional approaches suffer from severe prediction instability under real-world perturbations (baseline drift, sensor noise, and sampling defects), manifesting as erratic probability oscillations and fragmented decision boundaries vulnerable to samples with perturbations. **Right**: Our framework establishes hierarchical stability through signal morphology preservation (purification pipeline), prediction consistency anchoring, and perturbation-invariant decision manifolds.

during differential diagnosis. This cross-resolution reasoning process is not adequately captured by current multi-scale models, which often rely on naive feature concatenation rather than structured interaction Lawhern et al. (2018); Shan et al. (2022). Second, prevailing training paradigms prioritize accuracy on clean data at the expense of clinical robustness, leading to "silent failures" where minor input perturbations induce disproportionate prediction shifts Xu et al. (2021); Song et al. (2024). This performance–reliability mismatch poses a significant barrier to real-world deployment.

To address these challenges, we propose SM framework that synergistically co-designs a physiologically-inspired architecture with a principled, stability-constrained optimization strategy. Our contributions are threefold:

- We introduce the Stability-Aware Hierarchical Spatial Modulation (SHSM) module, a novel architecture that mimics clinical diagnostic reasoning. It dynamically separates salient, biomarker-correlated channels for focused sparse attention from residual signals that are processed by morphology-preserving convolutions. This allows the model to amplify critical diagnostic patterns while maintaining global waveform integrity.

- We propose a multifaceted stability optimization strategy that enforces diagnostic consistency. This strategy co-trains the model to maintain high accuracy on clean data, achieve robust performance against adversarial perturbations, enforce output consistency between original and perturbed inputs, and preserve diagnostic knowledge via self-distillation.

- We conduct extensive evaluations across four medical time series datasets, demonstrating that SM significantly outperforms 11 state-of-the-art baselines in both subject-dependent and subject-independent settings. Our results validate the effectiveness of our synergistic design in bridging the gap between laboratory performance and clinical reliability.

Together, these contributions establish a principled framework for enhancing diagnostic stability that offers both mechanistic insights and practical robustness improvements without requiring specialized hardware. By bridging the critical gap between laboratory performance and clinical reliability, this work lays the foundation for trustworthy medical AI systems.

## 2 RELATED WORK

### 2.1 MEDICAL TIME SERIES ANALYSIS.

Medical time series (MedTS), encompassing electrophysiological signals like EEG, ECG, and EMG, play a pivotal role in clinical diagnostics and neuroengineering Liu et al. (2021); Xiao et al. (2023). Unlike generic time series analysis primarily focused on forecasting, MedTS analysis prioritizes *signal decoding*—extracting disease-specific biomarkers from transient patterns across multi-scale temporal hierarchies (e.g., ECG's P-QRS-T complexes spanning milliseconds to minutes) Wang et al. (2023); Kiyasseh et al. (2021). Early approaches relied on handcrafted spectral features (e.g., inter-band power ratios Fahimi et al. (2017)) or shallow CNNs Lawhern et al. (2018) but struggled with real-world artifacts such as motion-induced noise and session-level variability. Recent advances integrate temporal-convolutional networks (TCNs) with attention mechanisms Song et al. (2022); Wang et al. (2024a) to model hierarchical dependencies. Notably, Medformer Wang et al. (2024b) introduced cross-channel multi-granularity patching and router-mediated attention, achieving state-of-the-art performance on benchmark datasets for tasks like arrhythmia detection.

However, prevailing methods inadequately address two key MedTS-specific challenges critical for real-world deployment: (1) *Hierarchical fragility*—existing architectures tend to flatten multi-scale temporal interactions, rendering biomarker representations vulnerable to localized noise; and (2) *Device heterogeneity*—models often overfit to acquisition-specific artifacts (e.g., electrode impedance variations in EEG), degrading performance across different clinical environments. Our work directly addresses these gaps through stability-driven architectural innovation and comprehensive adversarial optimization, establishing a new paradigm for *deployable* MedTS analysis that harmonizes accuracy, invariance, and efficiency.

### 2.2 ROBUST TIME SERIES ANALYSIS.

Robust time-series analysis has historically evolved along two main trajectories: statistical autoregressive models with noise suppression Franke (1984); Li et al. (2023) and deep learning with stability-driven training Cheng et al. (2023); Yu et al. (2024). The latter often employs techniques such as adversarial training, consistency regularization, and knowledge distillation, which have been successfully applied in fields like computer vision and semi-supervised learning. Early deep learning efforts enhanced models via data perturbations Wen et al. (2021) or specialized loss functions Guo et al. (2016), yet they struggled with the complex, nonlinear, and pathology-specific patterns inherent in medical signals. Modern deep networks improve generalization through adversarial training Cheng et al. (2023), model ensembles Krstanovic & Paulheim (2017), or decomposition architectures Yu et al. (2024).

Despite these advances, existing techniques exhibit critical shortcomings when applied to MedTS diagnosis: (1) They often collapse hierarchical temporal interactions into flat representations, losing valuable multi-scale clinical context; (2) Stability mechanisms (e.g., LSS Zhang et al. (2023)) primarily focus on generic noise, neglecting MedTS-specific artifacts; and (3) Most frameworks Queen et al. (2024); Zhou (2020) prioritize lab-based accuracy at the expense of clinical deployability. Our approach is distinct in that it does not merely apply these known stability techniques in isolation. Instead, we propose a holistic framework where a novel, physiologically-inspired architecture (SHSM) is co-designed with a multifaceted optimization objective. This synergy is crucial for achieving robustness that is tailored to the hierarchical and noisy nature of MedTS. Crucially, we redefine robustness not merely as resistance to generic noise but as decision invariance across perturbations—a fundamental prerequisite for trustworthy medical AI.

## 3 PROBLEM FORMULATION AND TASK CHARACTERIZATION

Medical time series (MedTS) analysis for disease diagnosis must reconcile two conflicting realities: the hierarchical temporal organization of physiological patterns (e.g., EEG rhythms, ECG waveform morphologies) and the pervasive noise artifacts in real-world clinical recordings. Although modalities such as EEG and ECG capture critical biomarkers, their diagnostic utility is compromised by inter-subject variability, non-stationary noise (e.g., motion artifacts in EEG, baseline wander in ECG),

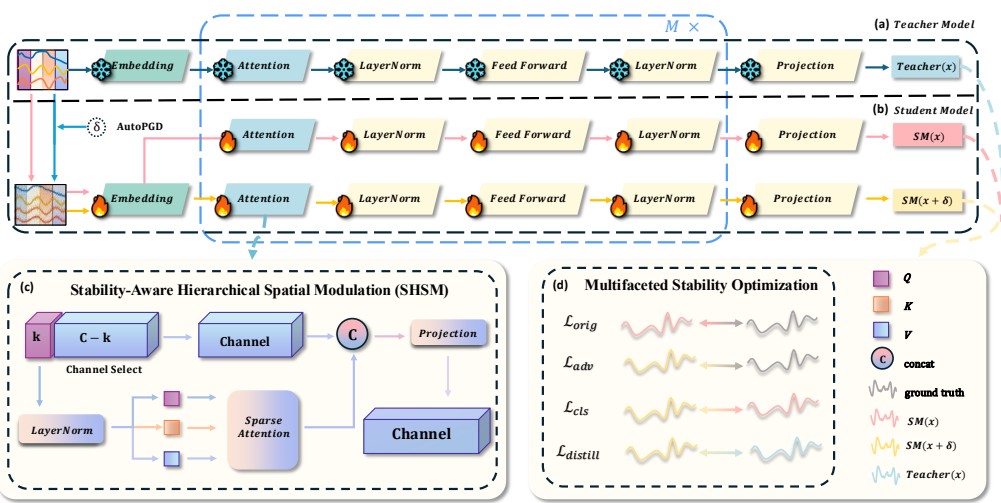

Figure 2: Architecture of the proposed Stability-driven Medical time series model (SM). Our framework enhances the architectural foundation with stability objectives and a novel processing module. (a) The **Teacher Model** (pre-trained and frozen) processes the clean input $x$ to provide stable targets for knowledge distillation. (b) The **Student Model (SM)** is trained to process both clean inputs $x$ and adversarially perturbed inputs $x+\delta$. Perturbations $\delta$ are generated using AutoPGD to maximize instability. (c) **Multifaceted Stability Optimization** employs four loss components: $\mathcal{L}_{orig}$ ensures performance on clean data; $\mathcal{L}_{adv}$ enforces robustness against adversarial perturbations; $\mathcal{L}_{cls}$ promotes output consistency between SM($x$) and SM($x + \delta$); and $\mathcal{L}_{distill}$ transfers stable knowledge from the teacher model to the perturbed student output SM($x + \delta$). (d) The **Stability-Aware Hierarchical Spatial Modulation (SHSM)** module replaces standard attention within SM layers. It performs channel selection based on feature energy ($k$ salient channels), processes these using Sparse Attention, and integrates them with morphology-preserved features from the remaining channels ($C - k$) via concatenation (C) and a final Projection, leading to a stability-aware representation.

and device-specific signal distortions. Together, these factors create a substantial gap between performance on controlled benchmarks and diagnostic consistency in clinical environments.

**Diagnostic Task Scope.** The core task is to map fixed-length segments of physiological signals to disease labels while meeting two clinical imperatives: **1) Multi-Scale Temporal Integration**: Local morphological features (e.g., ST-segment deviations in ECG) must be coherently synthesized with global trend dynamics (e.g., seizure evolution in EEG), mirroring the hierarchical reasoning clinicians employ. **2) Subject-Independent Generalization**: To avoid overfitting to subject-specific noise, training and evaluation must enforce strict separation of subjects.

**Operational Constraints.** **1) Input**: Fixed-length biosignal segments $x \in \mathbb{R}^{T \times C}$ derived from raw recordings, where $T$ is the temporal window and $C$ the sensor channels. **2) Output**: Multi-label vector $y \in \{0, 1\}^K$ indicating presence/absence of $K$ pathologies. **3) Critical Protocol**: Subject-level data partitioning ensures no overlap between training ($\mathcal{S}_{\text{train}}$) and test ($\mathcal{S}_{\text{test}}$) subjects.

**Key Limitations of Current Paradigms.** **1) Fragmented Multi-Scale Analysis**: Existing architectures treat different temporal resolutions independently and lack structured mechanisms for cross-scale feature interaction. Although Medformer Wang et al. (2024b) introduces cross-channel, multi-granularity patching, it still falls short of explicitly modeling hierarchical diagnostic dependencies. **2) Noise-Induced Output Instability**: Small input perturbations (e.g., electrode repositioning) can disproportionately affect model outputs, leading to brittle predictions. **3) Subject-Specific Overfitting**: Models tend to memorize idiosyncratic noise characteristics of training subjects, degrading performance on unseen cohorts. While Wang et al. Wang et al. (2024b) identify these issues, no

existing method provides an end-to-end solution addressing all these points comprehensively within a stability-driven framework.

These limitations motivate a fundamental shift from accuracy-centric training to stability-aware learning. In Section 5, we introduce our methodology, which integrates physiologically grounded hierarchical processing with stability constraints derived from clinical diagnostic principles.

## 4 PRELIMINARY: MEDFORMER AS A BASE ARCHITECTURE

Our proposed model, SM, builds upon the architectural foundation of Medformer Wang et al. (2024b). We adopt its effective Cross-Channel Multi-Granularity Patching scheme to handle the multi-scale nature of MedTS but replace its core attention mechanism with our proposed SHSM module. We briefly review the patching mechanism here.

**Cross-Channel Multi-Granularity Patching.** For an input $x_{\text{in}} \in \mathbb{R}^{T \times C}$, Medformer produces $n$ granularity-specific embeddings via: 1) **Multi-scale Segmentation**: For patch lengths $\{L_i\}_{i=1}^n$, divide $x_{\text{in}}$ into $N_i = \lceil T/L_i \rceil$ non-overlapping patches $x_p^{(i)} \in \mathbb{R}^{N_i \times (L_i \cdot C)}$. 2) **Projection & Augmentation**: Apply linear projection $x_e^{(i)} = x_p^{(i)} W^{(i)}$ followed by stochastic augmentation $\widetilde{x}_e^{(i)}$. 3) **Hierarchical Embedding**: Combine positional and granularity-specific encodings: $x^{(i)} = \widetilde{x}_e^{(i)} + W_{\text{pos}}[1 : N_i] + W_{\text{gr}}^{(i)}$, where $W_{\text{pos}}$ and $W_{\text{gr}}^{(i)}$ denote positional and granularity embeddings.

These patch embeddings $\{x^{(i)}\}_{i=1}^n$ for each granularity are then processed by a stack of transformer-style encoder layers. In the original Medformer, these layers use a router-mediated attention mechanism. In SM, we replace this mechanism with our SHSM module, as detailed in Section 5.1. The final representation $h$ is formed by concatenating the updated patch embeddings from all granularities, which is then used for downstream classification.

## 5 METHODOLOGY

Our methodology enhances diagnostic reliability through two core innovations: a stability-driven architectural modification to hierarchical feature interaction and a multifaceted stability optimization paradigm. These components work synergistically to address the clinical challenges of noise resilience, inter-subject generalization, and decision consistency. The architecture of the proposed **SM** is illustrated in Figure 2.

### 5.1 STABILITY-AWARE HIERARCHICAL SPATIAL MODULATION (SHSM) (D)

The SHSM module replaces the standard self-attention mechanism within each encoder layer of the base architecture. It is designed to selectively process information, inspired by how clinicians focus on high-yield diagnostic signals while maintaining awareness of the overall context. For an input feature map $x^{(i)} \in \mathbb{R}^{N_i \times D}$ corresponding to the $i$-th granularity (with $N_i$ patches and feature dimension $D$), SHSM operates in three steps:

**1. Energy-based Channel Selection.** The module first identifies the most informative channels. The intuition is that channels carrying critical diagnostic information (e.g., a QRS complex in ECG) often exhibit higher signal energy. We compute the L2-norm for each of the $C$ original signal channels across the temporal dimension of the input features to quantify this energy. We then select the top-$k$ channels for focused attention and designate the rest as residual channels for context preservation.

$$x_{\text{att}}^{(i)}, x_{\text{res}}^{(i)} = \text{ChannelSelect}(x^{(i)}, \text{TopK}(\|x^{(i)}\|_{2,\text{dim}=1}, k)), \quad k = \lfloor C/\alpha \rfloor. \tag{1}$$

Here, ChannelSelect$(\cdot, \text{indices})$ splits the input features into two groups: $x_{\text{att}}^{(i)}$ containing the $k$ salient channels and $x_{\text{res}}^{(i)}$ containing the remaining $C - k$ channels. The hyperparameter $\alpha$ controls the selection ratio, which is tuned on a validation set.

**2. Sparse Attention on Salient Channels.** The selected salient channels $x_{\text{att}}^{(i)}$ are processed by a single-head attention mechanism. To further emulate a clinician's focus, we employ a Gumbel-softmax based sparse attention mechanism Shan et al. (2022). This encourages the model to attend to

only the most critical temporal segments within these already-salient channels, promoting robustness by filtering out less relevant or noisy information.

$$\widetilde{\boldsymbol{x}}_{\text{att}}^{(i)} = \text{LayerNorm}\left(\text{SparseAttention}(\boldsymbol{x}_{\text{att}}^{(i)}\boldsymbol{W}^Q, \boldsymbol{x}_{\text{att}}^{(i)}\boldsymbol{W}^K, \boldsymbol{x}_{\text{att}}^{(i)}\boldsymbol{W}^V)\right), \tag{2}$$

3. Morphology Preservation and Fusion. The residual channels $\boldsymbol{x}_{\text{res}}^{(i)}$, which represent the global context and baseline trends, are processed by a lightweight depth-wise convolution $\mathcal{F}_{\text{conv}}$. This operation preserves their temporal morphology without the complexity of a full attention mechanism. Finally, the outputs from both pathways are fused to create a comprehensive, stability-aware representation.

$$\boldsymbol{x}_{\text{out}}^{(i)} = \text{Linear}(\text{Concat}(\widetilde{\boldsymbol{x}}_{\text{att}}^{(i)}, \boldsymbol{x}_{\text{res}}^{(i)})) + \mathcal{F}_{\text{conv}}(\boldsymbol{x}_{\text{res}}^{(i)}), \tag{3}$$

This fusion combines the sparsely-attended salient features with the morphology-preserved contextual features, providing a robust representation for the subsequent layer.

## 5.2 MULTIFACETED STABILITY OPTIMIZATION (C)

To enforce diagnostic stability, we train SM using an adversarial optimization strategy with four distinct loss components. This approach draws inspiration from established techniques in adversarial robustness, semi-supervised learning, and knowledge distillation. Given an input $\boldsymbol{x}$ with ground truth label $\boldsymbol{y}$, we generate an adversarial perturbation $\delta$ using Auto Projected Gradient Descent (AutoPGD) Croce & Hein (2020):

$$\delta^* = \arg \max_{\|\delta\|_\infty \leq \epsilon} \mathcal{L}_{\text{pert-obj}}(\text{SM}(\boldsymbol{x}+\delta), \boldsymbol{y}; \theta), \tag{4}$$

where $\mathcal{L}_{\text{pert-obj}}$ is the perturbation-generating loss (typically cross-entropy), $\theta$ are the model parameters, and $\epsilon$ is the perturbation budget. The model is then updated by minimizing a total loss function:

$$\min_\theta \mathbb{E}_{(\boldsymbol{x},\boldsymbol{y})\sim\mathcal{D}} \left[\mathcal{L}_{orig} + \lambda_{adv}\mathcal{L}_{adv} + \lambda_{cls}\mathcal{L}_{cls} + \lambda_{distill}\mathcal{L}_{distill}\right], \tag{5}$$

where $\mathcal{D}$ is the data distribution, and $\lambda_i$ are weighting hyperparameters. The four loss components are: 1) Clean Data Classification Loss ($\mathcal{L}_{orig}$): The standard cross-entropy loss on the original input $\boldsymbol{x}$, ensuring high performance on clean data.

$$\mathcal{L}_{orig} = \text{CrossEntropy}(\text{SM}(\boldsymbol{x}), \boldsymbol{y}). \tag{6}$$

2) Adversarial Classification Loss ($\mathcal{L}_{adv}$): The cross-entropy loss on the perturbed input $\boldsymbol{x}+\delta$, directly encouraging robustness against adversarial examples.

$$\mathcal{L}_{adv} = \text{CrossEntropy}(\text{SM}(\boldsymbol{x}+\delta), \boldsymbol{y}). \tag{7}$$

3) Output Consistency Loss ($\mathcal{L}_{cls}$): This loss, inspired by consistency regularization methods, penalizes divergence between the model's output distributions on clean and perturbed inputs. We use Mean Squared Error (MSE) for its simplicity and effectiveness in penalizing large deviations in probability scores.

$$\mathcal{L}_{cls} = \|\text{Softmax}(\text{SM}(\boldsymbol{x})) - \text{Softmax}(\text{SM}(\boldsymbol{x}+\delta))\|_2^2. \tag{8}$$

4) Knowledge Distillation Loss ($\mathcal{L}_{distill}$): We use a pre-trained, frozen teacher model (Teacher) to guide the student model's training under perturbation, a common technique for stabilizing training. This loss aligns the student's output on the perturbed input with the teacher's stable output on the clean input, using Kullback-Leibler (KL) divergence.

$$\mathcal{L}_{distill} = D_{\text{KL}}\left(\text{Softmax}(\text{Teacher}(\boldsymbol{x})/\tau) \,\|\, \text{Softmax}(\text{SM}(\boldsymbol{x}+\delta)/\tau)\right), \tag{9}$$

where $\tau$ is a temperature parameter that softens the probability distributions. The teacher is a copy of the model pre-trained on clean data.

This multifaceted objective ensures the model learns not just to classify correctly, but to do so in a stable and consistent manner, which is critical for clinical deployment.

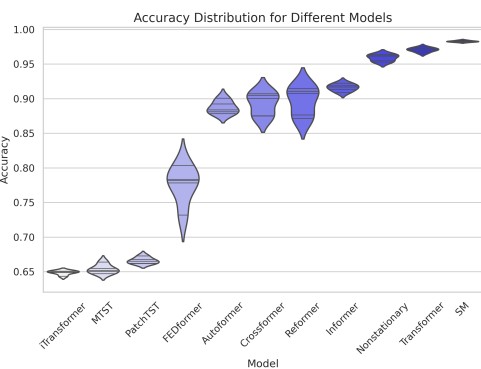
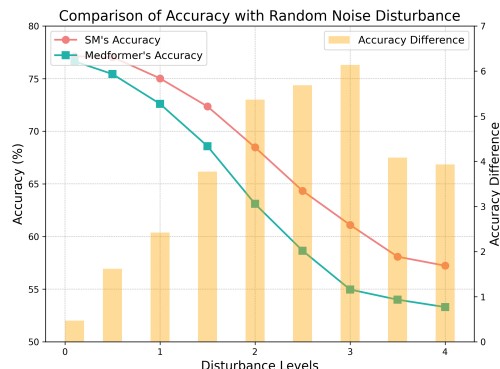

Figure 3: Violin plots showing the distribution of Accuracy for different models across 5 random seeds on the APAVA dataset. The x-axis represents the model name, and the y-axis shows the accuracy (%). A narrower violin shape, particularly at the extremes, indicates lower variability in accuracy across different random seeds, implying greater robustness. The vertical position of the violin's thick bar represents the median accuracy, with a higher position indicating better typical performance.

Figure 4: Evaluation of Model Accuracy under Random Noise Perturbation on the APAVA Dataset. The x-axis denotes the level of random noise perturbation applied to the input data, expressed as a percentage (%). The left y-axis displays the model's accuracy (%), shown as a line graph. The right y-axis presents the corresponding accuracy difference (%), representing the absolute reduction in accuracy compared to the model's performance on unperturbed data, depicted by the bar chart.

## 6 EXPERIMENTS

**Datasets.** We evaluate our model on four medical time series datasets: three EEG datasets (APAVA Escudero et al. (2006), TDBrain van Dijk et al. (2022), ADFTD Miltiadous et al. (2023b)) and one ECG dataset (PTB PhysioBank (2000)). Following prior work Wang et al. (2024b), APAVA, TDBrain, and PTB use a subject-independent split, while ADFTD is assessed using a subject-dependent split. Details are in Appendix C.1.

**Baselines.** We compare SM against 11 Transformer-based models for time series analysis: Autoformer Wu et al. (2021), Crossformer Zhang & Yan (2022), FEDformer Zhou et al. (2022), Informer Zhou et al. (2021), iTransformer Liu et al. (2024), MTST Zhang et al. (2024), Nonformer Liu et al. (2022), PatchTST Nie et al. (2023), Reformer Kitaev et al. (2019), a vanilla Transformer Vaswani et al. (2017), and Medformer Wang et al. (2024b).

### 6.1 COMPARISON TO STATE-OF-THE-ART METHODS

Table 1 shows a comprehensive comparison. On the APAVA and ADFTD datasets, SM achieves state-of-the-art (SOTA) performance across all six metrics. On TDBrain and PTB, SM achieves superior performance on multiple critical metrics, demonstrating its strong overall capability.

### 6.2 STABILITY EVALUATION

**Stability to Random Seeds:** We first evaluate stability against variations in random seeds. The violin plots in Figure 3 show that SM not only achieves SOTA performance but also exhibits the narrowest distribution, indicating high resilience to initialization variance.

**Robustness to Noise Perturbations:** To evaluate robustness, we injected Gaussian random noise at varying levels into the test data. While Gaussian noise is a simplification of complex clinical artifacts, it serves as a standard benchmark for assessing a model's general stability against unforeseen perturbations. Figure 4 and Table 2 show that SM maintains significantly higher accuracy and precision than Medformer as noise increases. This highlights its improved reliability, which is critical for preventing false positives and ensuring correct diagnoses in noisy clinical environments.

Table 1: **Overall Performance Comparison on Multiple Datasets.** The best performance for each metric and dataset is indicated in **bold**. Note that the APAVA, TDBrain, and PTB datasets are assessed using a subject-independent protocol, whereas the ADFTD dataset employs a subject-dependent split.

| Datasets | Models | Accuracy | Precision | Recall | F1 score | AUROC | AUPRC |
|---|---|---|---|---|---|---|---|
| APAVA
(2-Classes) | Autoformer | $68.64_{\pm1.82}$ | $68.48_{\pm2.10}$ | $68.64_{\pm1.82}$ | $68.07_{\pm1.94}$ | $75.94_{\pm3.61}$ | $74.38_{\pm4.05}$ |
| | Crossformer | $73.77_{\pm1.95}$ | $79.29_{\pm4.36}$ | $68.86_{\pm1.70}$ | $68.86_{\pm1.70}$ | $72.39_{\pm3.33}$ | $72.05_{\pm3.65}$ |
| | FEDformer | $74.94_{\pm2.15}$ | $74.94_{\pm2.15}$ | $73.51_{\pm3.35}$ | $73.51_{\pm3.35}$ | $83.72_{\pm1.97}$ | $82.94_{\pm2.37}$ |
| | Informer | $73.11_{\pm4.40}$ | $75.17_{\pm6.06}$ | $69.17_{\pm4.56}$ | $69.47_{\pm5.06}$ | $70.46_{\pm4.91}$ | $70.75_{\pm5.27}$ |
| | iTransformer | $74.55_{\pm1.66}$ | $74.77_{\pm1.20}$ | $71.76_{\pm1.72}$ | $72.30_{\pm1.79}$ | $85.59_{\pm1.55}$ | $84.39_{\pm1.57}$ |
| | MTST | $71.14_{\pm1.59}$ | $79.30_{\pm0.97}$ | $65.27_{\pm1.22}$ | $64.01_{\pm3.16}$ | $68.87_{\pm2.34}$ | $71.06_{\pm1.60}$ |
| | Nonformer | $71.89_{\pm3.81}$ | $71.80_{\pm4.58}$ | $69.44_{\pm3.56}$ | $69.74_{\pm3.84}$ | $70.55_{\pm2.96}$ | $70.78_{\pm4.08}$ |
| | PatchTST | $67.03_{\pm1.65}$ | $78.76_{\pm1.28}$ | $59.91_{\pm2.02}$ | $55.97_{\pm3.10}$ | $65.65_{\pm0.28}$ | $67.99_{\pm0.76}$ |
| | Reformer | $78.70_{\pm2.00}$ | $82.50_{\pm3.95}$ | $75.00_{\pm4.61}$ | $75.93_{\pm4.82}$ | $73.94_{\pm4.14}$ | $76.04_{\pm4.11}$ |
| | Transformer | $76.30_{\pm4.72}$ | $77.64_{\pm4.95}$ | $73.09_{\pm5.01}$ | $73.75_{\pm4.53}$ | $72.50_{\pm6.60}$ | $73.23_{\pm7.60}$ |
| | Medformer | $76.99_{\pm2.72}$ | $77.58_{\pm4.09}$ | $74.82_{\pm1.83}$ | $75.37_{\pm2.22}$ | $82.93_{\pm2.31}$ | $83.70_{\pm2.08}$ |
| | SM(Ours) | $\mathbf{79.34_{\pm1.17}}$ | $\mathbf{83.27_{\pm1.60}}$ | $\mathbf{75.59_{\pm1.39}}$ | $\mathbf{76.56_{\pm1.50}}$ | $\mathbf{85.48_{\pm2.59}}$ | $\mathbf{85.35_{\pm2.94}}$ |
| TDBrain
(2-Classes) | Autoformer | $87.33_{\pm3.79}$ | $88.06_{\pm3.56}$ | $87.33_{\pm3.79}$ | $87.26_{\pm3.84}$ | $93.81_{\pm4.22}$ | $93.32_{\pm4.42}$ |
| | Crossformer | $81.56_{\pm2.19}$ | $81.97_{\pm4.25}$ | $81.56_{\pm2.19}$ | $81.50_{\pm2.20}$ | $91.20_{\pm4.17}$ | $91.51_{\pm4.17}$ |
| | FEDformer | $78.13_{\pm4.98}$ | $78.52_{\pm4.91}$ | $78.13_{\pm4.98}$ | $78.04_{\pm2.01}$ | $86.56_{\pm1.86}$ | $86.48_{\pm1.99}$ |
| | Informer | $89.02_{\pm2.50}$ | $89.43_{\pm2.14}$ | $89.02_{\pm2.50}$ | $88.98_{\pm2.54}$ | $\mathbf{96.64_{\pm0.68}}$ | $\mathbf{96.75_{\pm0.63}}$ |
| | iTransformer | $74.67_{\pm1.06}$ | $74.71_{\pm1.06}$ | $74.67_{\pm1.06}$ | $74.65_{\pm1.06}$ | $83.37_{\pm1.14}$ | $83.73_{\pm1.27}$ |
| | MTST | $76.96_{\pm3.76}$ | $77.24_{\pm3.59}$ | $76.96_{\pm3.76}$ | $76.88_{\pm3.83}$ | $85.27_{\pm4.46}$ | $82.81_{\pm4.65}$ |
| | Nonformer | $87.88_{\pm4.28}$ | $88.86_{\pm4.18}$ | $87.88_{\pm4.28}$ | $87.78_{\pm4.26}$ | $97.05_{\pm0.68}$ | $96.99_{\pm0.68}$ |
| | PatchTST | $79.25_{\pm3.79}$ | $79.60_{\pm4.09}$ | $79.25_{\pm3.79}$ | $79.20_{\pm3.77}$ | $87.95_{\pm4.96}$ | $86.36_{\pm6.67}$ |
| | Reformer | $87.92_{\pm4.01}$ | $88.64_{\pm4.14}$ | $87.92_{\pm4.01}$ | $87.85_{\pm4.20}$ | $96.30_{\pm5.04}$ | $96.40_{\pm4.45}$ |
| | Transformer | $87.17_{\pm4.67}$ | $87.99_{\pm4.68}$ | $87.17_{\pm4.67}$ | $87.10_{\pm4.68}$ | $96.28_{\pm9.92}$ | $96.34_{\pm8.11}$ |
| | Medformer | $88.08_{\pm0.43}$ | $88.19_{\pm0.44}$ | $88.08_{\pm0.43}$ | $88.07_{\pm0.43}$ | $95.69_{\pm0.20}$ | $95.65_{\pm0.16}$ |
| | SM(Ours) | $\mathbf{90.00_{\pm1.18}}$ | $\mathbf{90.12_{\pm1.08}}$ | $\mathbf{90.00_{\pm1.18}}$ | $\mathbf{90.00_{\pm1.20}}$ | $96.32_{\pm0.66}$ | $96.41_{\pm0.64}$ |
| ADFTD-Dep
(3-Classes) | Autoformer | $87.83_{\pm1.62}$ | $87.63_{\pm1.66}$ | $87.22_{\pm1.97}$ | $87.38_{\pm1.79}$ | $96.59_{\pm0.88}$ | $93.82_{\pm1.64}$ |
| | Crossformer | $89.35_{\pm1.32}$ | $89.00_{\pm1.44}$ | $88.79_{\pm1.37}$ | $88.88_{\pm1.40}$ | $97.52_{\pm0.58}$ | $95.45_{\pm1.03}$ |
| | FEDformer | $77.63_{\pm2.37}$ | $76.76_{\pm2.17}$ | $76.68_{\pm2.48}$ | $76.60_{\pm2.46}$ | $91.67_{\pm1.34}$ | $84.94_{\pm2.11}$ |
| | Informer | $90.93_{\pm0.90}$ | $90.74_{\pm0.71}$ | $90.50_{\pm1.14}$ | $90.60_{\pm0.94}$ | $98.19_{\pm0.27}$ | $96.51_{\pm0.49}$ |
| | iTransformer | $64.90_{\pm0.25}$ | $62.53_{\pm0.27}$ | $62.21_{\pm0.26}$ | $62.25_{\pm0.33}$ | $81.52_{\pm0.29}$ | $68.87_{\pm0.49}$ |
| | MTST | $65.08_{\pm0.69}$ | $63.85_{\pm0.80}$ | $62.71_{\pm0.64}$ | $63.03_{\pm0.58}$ | $81.36_{\pm0.56}$ | $69.34_{\pm0.89}$ |
| | Nonformer | $96.12_{\pm0.47}$ | $95.94_{\pm0.56}$ | $95.99_{\pm0.38}$ | $95.96_{\pm0.47}$ | $99.59_{\pm0.09}$ | $99.08_{\pm0.16}$ |
| | PatchTST | $66.26_{\pm0.40}$ | $65.08_{\pm0.41}$ | $64.97_{\pm0.51}$ | $64.95_{\pm0.42}$ | $83.07_{\pm0.45}$ | $71.70_{\pm0.61}$ |
| | Reformer | $91.51_{\pm1.75}$ | $91.15_{\pm1.79}$ | $91.65_{\pm1.56}$ | $91.14_{\pm1.83}$ | $98.85_{\pm0.35}$ | $97.88_{\pm0.60}$ |
| | Transformer | $97.00_{\pm0.43}$ | $96.87_{\pm0.53}$ | $96.86_{\pm0.36}$ | $96.86_{\pm0.44}$ | $99.75_{\pm0.04}$ | $99.42_{\pm0.07}$ |
| | Medformer | $97.48_{\pm0.16}$ | $97.57_{\pm0.18}$ | $97.51_{\pm0.18}$ | $97.48_{\pm0.17}$ | $99.57_{\pm0.02}$ | $99.45_{\pm0.04}$ |
| | SM(Ours) | $\mathbf{98.29_{\pm0.09}}$ | $\mathbf{98.21_{\pm0.10}}$ | $\mathbf{98.21_{\pm0.11}}$ | $\mathbf{98.21_{\pm0.09}}$ | $\mathbf{99.89_{\pm0.01}}$ | $\mathbf{99.78_{\pm0.03}}$ |
| PTB
(2-Classes) | Autoformer | $73.35_{\pm2.10}$ | $72.11_{\pm4.28}$ | $63.24_{\pm3.17}$ | $63.69_{\pm3.84}$ | $78.54_{\pm3.48}$ | $74.25_{\pm3.53}$ |
| | Crossformer | $80.17_{\pm3.79}$ | $85.04_{\pm4.18}$ | $71.25_{\pm4.29}$ | $72.75_{\pm4.72}$ | $88.55_{\pm4.35}$ | $87.31_{\pm4.32}$ |
| | FEDformer | $76.05_{\pm4.25}$ | $77.58_{\pm4.36}$ | $66.10_{\pm4.35}$ | $67.14_{\pm4.37}$ | $85.93_{\pm4.31}$ | $82.59_{\pm5.42}$ |
| | Informer | $78.69_{\pm1.68}$ | $82.87_{\pm4.10}$ | $69.19_{\pm4.90}$ | $70.84_{\pm3.47}$ | $92.09_{\pm5.03}$ | $90.02_{\pm0.60}$ |
| | iTransformer | $\mathbf{83.02_{\pm0.74}}$ | $\mathbf{88.19_{\pm8.86}}$ | $74.89_{\pm1.01}$ | $77.54_{\pm1.12}$ | $90.77_{\pm1.14}$ | $90.69_{\pm0.97}$ |
| | MTST | $76.59_{\pm4.19}$ | $79.88_{\pm4.19}$ | $66.31_{\pm4.29}$ | $67.38_{\pm4.37}$ | $86.86_{\pm4.27}$ | $83.75_{\pm4.28}$ |
| | Nonformer | $78.66_{\pm4.09}$ | $82.77_{\pm4.06}$ | $69.12_{\pm4.87}$ | $70.90_{\pm4.05}$ | $89.37_{\pm4.51}$ | $86.67_{\pm4.23}$ |
| | PatchTST | $74.74_{\pm4.10}$ | $76.94_{\pm4.15}$ | $63.89_{\pm4.20}$ | $64.36_{\pm4.38}$ | $88.79_{\pm4.09}$ | $83.39_{\pm4.18}$ |
| | Reformer | $77.96_{\pm2.13}$ | $81.72_{\pm4.16}$ | $68.20_{\pm4.16}$ | $69.65_{\pm4.39}$ | $91.13_{\pm4.74}$ | $88.42_{\pm4.30}$ |
| | Transformer | $77.37_{\pm1.02}$ | $81.84_{\pm4.07}$ | $67.14_{\pm4.18}$ | $68.47_{\pm4.19}$ | $90.08_{\pm4.76}$ | $87.22_{\pm4.68}$ |
| | Medformer | $79.84_{\pm1.62}$ | $87.17_{\pm0.54}$ | $69.89_{\pm2.60}$ | $71.82_{\pm3.01}$ | $\mathbf{93.20_{\pm0.72}}$ | $\mathbf{92.67_{\pm0.71}}$ |
| | SM(Ours) | $82.12_{\pm1.17}$ | $86.81_{\pm1.17}$ | $\mathbf{76.85_{\pm1.90}}$ | $\mathbf{77.82_{\pm1.68}}$ | $90.22_{\pm1.56}$ | $90.13_{\pm1.51}$ |

## 6.3 ABLATION STUDY

We conducted an extensive ablation study to validate the effectiveness of our framework's components, presented in Table 3. **Impact of Stability Optimization:** Applying our multifaceted stability optimization to the baseline Medformer significantly boosts its performance across all metrics. This demonstrates the general effectiveness of our training strategy in enhancing robustness. **Impact of SHSM Module:** Comparing the baseline Medformer with an SM model trained only with $\mathcal{L}_{orig}$ (row 1 vs. row 3) shows that our SHSM architecture alone provides a performance gain. More importantly, comparing the fully-equipped SM with the stability-optimized Medformer (row 2 vs. final row) reveals that the SHSM module provides a further, clear improvement. This confirms that our architectural innovation and optimization strategy are synergistic, and both contribute to the final

Table 2: Comparison of Accuracy and Precision for SM and Medformer across different disturbance levels in APAVA.

| Metric | | Disturbance Amplitude | | | | | | | | |
|---|---|---|---|---|---|---|---|---|---|---|
| | | 0.1 | 0.5 | 1.0 | 1.5 | 2.0 | 2.5 | 3.0 | 3.5 | 4.0 |
| SM | Accuracy | 77.12 | 77.05 | 75.02 | 72.35 | 68.47 | 64.35 | 61.11 | 58.09 | 57.23 |
| | Precision | 82.03 | 81.56 | 77.26 | 72.53 | 67.56 | 63.05 | 59.90 | 56.98 | 56.34 |
| Medformer | Accuracy | 76.65 | 75.43 | 72.60 | 68.58 | 63.10 | 58.66 | 54.97 | 54.00 | 53.30 |
| | Precision | 78.61 | 76.91 | 72.65 | 68.13 | 63.28 | 59.05 | 55.82 | 54.06 | 54.06 |
| Difference | Accuracy | 0.47 | 1.62 | 2.42 | 3.77 | 5.37 | 5.69 | 6.14 | 4.08 | 3.93 |
| | Precision | 3.42 | 4.65 | 4.61 | 4.40 | 4.28 | 4.00 | 4.08 | 2.09 | 2.27 |

Table 3: Results of the Ablation Study on the Impact of Different Components of SM on the APAVA dataset.

| Base Model | SHSM | Stability Opt. | Losses Used | Accuracy | Precision | Recall | F1 | AUROC | AUPRC |
|---|---|---|---|---|---|---|---|---|---|
| Medformer | | | $\mathcal{L}_{orig}$ | $76.99_{\pm 2.72}$ | $77.58_{\pm 4.09}$ | $74.82_{\pm 1.83}$ | $75.37_{\pm 2.22}$ | $82.93_{\pm 2.31}$ | $83.70_{\pm 2.08}$ |
| | | ✓ | $\mathcal{L}_{orig} + \mathcal{L}_{adv} + \mathcal{L}_{cls} + \mathcal{L}_{distill}$ | $78.21_{\pm 1.55}$ | $81.53_{\pm 1.80}$ | $74.95_{\pm 1.62}$ | $75.88_{\pm 1.71}$ | $84.12_{\pm 2.81}$ | $84.35_{\pm 2.77}$ |
| SM (Ours) | ✓ | | $\mathcal{L}_{orig}$ | $77.85_{\pm 2.10}$ | $80.15_{\pm 2.54}$ | $75.01_{\pm 2.33}$ | $75.99_{\pm 2.41}$ | $83.55_{\pm 2.60}$ | $83.98_{\pm 2.51}$ |
| | ✓ | ✓ | $\mathcal{L}_{orig} + \mathcal{L}_{cls} + \mathcal{L}_{distill}$ | $79.27_{\pm 2.29}$ | $81.36_{\pm 1.99}$ | $76.12_{\pm 2.75}$ | $76.99_{\pm 2.90}$ | $85.17_{\pm 2.15}$ | $85.44_{\pm 1.98}$ |
| | ✓ | ✓ | $\mathcal{L}_{orig} + \mathcal{L}_{adv} + \mathcal{L}_{distill}$ | $76.88_{\pm 4.51}$ | $80.60_{\pm 2.69}$ | $72.92_{\pm 5.81}$ | $73.13_{\pm 7.16}$ | $81.51_{\pm 4.50}$ | $81.67_{\pm 4.25}$ |
| | ✓ | ✓ | All Four | $\mathbf{79.34}_{\pm 1.17}$ | $\mathbf{83.27}_{\pm 1.60}$ | $\mathbf{75.59}_{\pm 1.39}$ | $\mathbf{76.56}_{\pm 1.50}$ | $\mathbf{85.48}_{\pm 2.59}$ | $\mathbf{85.35}_{\pm 2.94}$ |

Table 4: Few-shot Learning Performance with Different Data Proportions on APAVA.

| Model | Data Proportion | Accuracy | Precision | Recall | F1 score | AUROC | AUPRC |
|---|---|---|---|---|---|---|---|
| SM (Ours) | 90% | $79.82_{\pm 0.88}$ | $82.05_{\pm 1.21}$ | $76.77_{\pm 1.37}$ | $77.67_{\pm 1.30}$ | $83.39_{\pm 3.37}$ | $83.45_{\pm 4.06}$ |
| | 85% | $79.15_{\pm 0.48}$ | $82.40_{\pm 1.82}$ | $75.62_{\pm 0.47}$ | $76.56_{\pm 0.48}$ | $83.48_{\pm 2.03}$ | $83.61_{\pm 2.18}$ |
| | ... | ... | ... | ... | ... | ... | ... |
| | 30% | $75.64_{\pm 1.62}$ | $76.87_{\pm 2.41}$ | $72.44_{\pm 1.40}$ | $73.09_{\pm 1.53}$ | $81.44_{\pm 3.49}$ | $80.91_{\pm 3.67}$ |
| Medformer | 100% | $76.99_{\pm 2.72}$ | $77.58_{\pm 4.09}$ | $74.82_{\pm 1.83}$ | $75.37_{\pm 2.22}$ | $82.93_{\pm 2.31}$ | $83.70_{\pm 2.08}$ |
| | 50% | $72.15_{\pm 2.95}$ | $74.30_{\pm 3.11}$ | $68.91_{\pm 3.05}$ | $69.88_{\pm 3.15}$ | $78.05_{\pm 3.50}$ | $78.21_{\pm 3.44}$ |
| | 30% | $68.83_{\pm 3.51}$ | $70.12_{\pm 3.88}$ | $65.20_{\pm 4.01}$ | $66.03_{\pm 4.12}$ | $74.52_{\pm 4.13}$ | $75.01_{\pm 4.20}$ |
| SM (Ours) | 100% | $\mathbf{79.34}_{\pm 1.17}$ | $\mathbf{83.27}_{\pm 1.60}$ | $\mathbf{75.59}_{\pm 1.39}$ | $\mathbf{76.56}_{\pm 1.50}$ | $\mathbf{85.48}_{\pm 2.59}$ | $\mathbf{85.35}_{\pm 2.94}$ |

performance. **Impact of Loss Components:** The final rows show that removing either $\mathcal{L}_{adv}$ or other stability losses degrades performance, with the combination of all four losses yielding the best result. This confirms that each component of our multifaceted objective captures a unique and valuable aspect of clinical robustness.

## 6.4 FEW-SHOT ABILITY EVALUATION

Table 4 evaluates SM's performance under limited data conditions. The results show that SM maintains high performance even with significantly reduced training data. When trained on only 30% of the data, the performance drop is minimal. Compared to the baseline Medformer, which exhibits a much sharper decline in performance, SM's graceful degradation highlights its superior data efficiency and generalization ability, making it highly suitable for clinical scenarios where labeled data is often scarce.

## 7 CONCLUSION

In this work, we identified a critical reliability gap in deep learning for medical time series analysis, stemming from inadequate modeling of hierarchical physiological interactions and training paradigms that prioritize clean-data accuracy over clinical robustness. To address this, we proposed **SM**, a framework integrating a novel **S**tability-aware **H**ierarchical **S**patial **M**odulation (SHSM) module with a multifaceted stability optimization strategy. Extensive evaluations demonstrated SM's superior performance and enhanced robustness. Our key contribution is a synergistic framework where a physiologically-inspired architecture and a comprehensive stability-driven training objective work in concert to bridge the laboratory-to-clinic gap. This work lays a foundation for more trustworthy and deployable AI systems in medical time series analysis.

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

## A  LIMITATIONS AND FUTURE WORK

Despite the promising results demonstrated by SM, we acknowledge several limitations that open avenues for future research.

- **Generalizability Across Tasks and Modalities:** Our evaluation focused on classification tasks across four specific datasets. Future work should validate SM's principles on a broader spectrum of medical time series (e.g., EMG, ECoG) and diagnostic tasks (e.g., anomaly detection, forecasting) to fully assess its generalizability.

Table 5: Dataset statistics used in our experiments.

| Datasets | #-Subject | #-Sample | #-Class | #-Channel | Sampling Rate | Modality | File Size |
|---|---|---|---|---|---|---|---|
| APAVA Escudero et al. (2006) | 23 | 5,967 | 2 | 16 | 256Hz | EEG | 186MB |
| TDBrain van Dijk et al. (2022) | 72 | 6,240 | 2 | 33 | 256Hz | EEG | 571MB |
| ADFTD Miltiadous et al. (2023b;a) | 88 | 69,752 | 3 | 19 | 256Hz | EEG | 2.52GB |
| PTB PhysioBank (2000) | 198 | 64,356 | 2 | 15 | 250Hz | ECG | 2.15GB |

- **Complexity of Clinical Artifacts:** While we evaluated robustness against Gaussian noise and standard adversarial attacks, real-world clinical artifacts can be more complex and structured (e.g., motion-induced spikes, baseline wander). Future work should incorporate more realistic, modality-specific noise models to further close the gap to clinical deployment.

- **Interpretability:** While the SHSM module is clinically inspired, a deeper analysis of the features it learns could provide valuable insights and increase clinical trust. Developing methods to visualize the specific physiological patterns the model attends to would be a valuable extension.

- **Computational Cost:** The adversarial training component, while crucial for robustness, increases training time. Exploring more efficient stability-inducing regularization methods that maintain robustness while reducing computational overhead could be beneficial for large-scale deployments. We provide a detailed cost analysis in Appendix D.4.

## B  BROADER IMPACT AND ETHICAL CONSIDERATIONS

The development of robust AI for medical diagnosis has significant potential for positive societal impact by improving diagnostic accuracy, reducing clinician workload, and enabling access to care. Our work, by focusing on the stability and reliability of these models, aims to contribute to the safe and effective translation of AI from the lab to the clinic.

However, this research also carries ethical responsibilities that must be addressed.

- **Over-reliance and Automation Bias:** An overly trusted AI system could lead clinicians to accept incorrect suggestions, potentially leading to diagnostic errors. We stress that SM should be deployed as a **decision-support tool** to assist, not replace, qualified medical professionals.

- **Data Bias and Health Equity:** The model's performance is contingent on the training data. If datasets are not diverse in terms of demographics, pathologies, and acquisition hardware, the model may perpetuate existing health disparities.

- **Accountability:** In the event of an AI-involved diagnostic error, determining accountability is complex. Clear regulatory frameworks are needed to manage the responsibilities of developers, healthcare providers, and institutions.

We encourage future work to actively address these challenges, particularly by focusing on fairness audits and validating performance across diverse, multi-center clinical datasets.

## C  EXPERIMENTAL SETUP DETAILS

### C.1  DATASETS AND PREPROCESSING

The characteristics of the datasets used in this study are summarized in Table 5.

The **APAVA** dataset comprises 23 subjects (12 Alzheimer's patients and 11 healthy controls). Each trial is divided into overlapping 1-second samples, resulting in 5,967 samples for binary classification. A subject-independent split is used.

The **TDBrain** dataset contains EEG recordings from 72 subjects for a binary classification task. A total of 6,240 samples are generated from 1-second, non-overlapping segments, with a subject-independent split.

The **ADFTD** dataset includes 88 subjects (36 Alzheimer's, 23 FTD, and 29 healthy controls) for 3-class classification. Non-overlapping 1-second segments yield 69,752 samples. Data partitioning is performed using both subject-dependent and independent splits.

The **PTB** dataset contains ECG data from 198 subjects for arrhythmia classification. Each sample represents a single heartbeat. A subject-independent approach is used to create the splits, totaling 64,356 samples.

## C.2 BASELINE MODELS

We compare SM against 11 Transformer-based time series models, implemented within the unified Time-Series-Library Wu et al. (2023) to ensure a fair comparison.

- **Autoformer** Wu et al. (2021) introduces a decomposition architecture with an Auto-Correlation mechanism for modeling temporal dependencies.
- **Crossformer** Zhang & Yan (2022) utilizes a cross-dimension self-attention mechanism to capture dependencies across different dimensions of multivariate time series.
- **FEDformer** Zhou et al. (2022) combines seasonal-trend decomposition with a frequency-enhanced transformer, using Fourier transforms to model frequency components.
- **Informer** Zhou et al. (2021) addresses long sequence forecasting with a ProbSparse self-attention mechanism to reduce complexity.
- **iTransformer** Liu et al. (2024) incorporates inter- and intra-variable attention mechanisms to capture complex temporal and variable dependencies.
- **MTST** Zhang et al. (2024) employs a multi-resolution approach, segmenting time series into patches of varying lengths to capture periodic components at different scales.
- **Nonformer** Liu et al. (2022) introduces a non-linear attention mechanism to adapt to the inherent non-linearity of time series data.
- **PatchTST** Nie et al. (2023) divides time series into fixed-length patches, treating each patch as a token to focus on local patterns.
- **Reformer** Kitaev et al. (2019) optimizes transformers through reversible layers and locality-sensitive hashing (LSH) attention to reduce memory usage.
- **Transformer** Vaswani et al. (2017) is the vanilla Transformer model applied to time series data.
- **Medformer** Wang et al. (2024b) is specifically tailored for medical time-series, integrating multi-granularity patching with a transformer architecture.

## C.3 IMPLEMENTATION DETAILS

All experiments were conducted on servers equipped with NVIDIA RTX 4090 and A800 GPUs. We report the mean and standard deviation across five random seeds (41-45) on fixed data splits. Performance is evaluated using six standard metrics: accuracy, precision, recall, F1 score, AUROC, and AUPRC (all macro-averaged).

Key hyperparameters for training our proposed SM model are detailed in Table 6. The teacher model used for distillation was a Medformer model pre-trained on the clean training data of each respective dataset.

The adversarial training for all models utilizes AutoPGD. The key parameters used in the AutoPGD procedure are detailed in our ablation study in Appendix D.1.

# D ADDITIONAL RESULTS AND ANALYSIS

## D.1 ABLATION STUDY ON ADVERSARIAL PERTURBATION STRATEGY

To evaluate the impact of the adversarial perturbation strategy, we conducted an ablation study on the key parameters of the AutoPGD method on the APAVA dataset. The results are presented in

Table 6: Key Hyperparameters for Training SM.

| Hyperparameter | Value |
|---|---|
| Optimizer | AdamW |
| Learning Rate | 1e-4 |
| Weight Decay | 1e-5 |
| Batch Size | 64 |
| Number of Epochs | 100 |
| Scheduler | Cosine Annealing |
| **Stability Optimization Hyperparameters** | |
| Adversarial Perturbation Budget ($\epsilon$) | 0.01 |
| Adversarial Loss Weight ($\lambda_{adv}$) | 1.0 |
| Consistency Loss Weight ($\lambda_{cls}$) | 0.5 |
| Distillation Loss Weight ($\lambda_{distill}$) | 0.5 |
| Distillation Temperature ($\tau$) | 2.0 |
| SHSM Channel Selection Ratio ($\alpha$) | 4.67 |

Table 7: Ablation study results for SM under different AutoPGD hyperparameter settings on APAVA.

| Parameter | Value | Accuracy | Precision | Recall | F1 | AUROC | AUPRC |
|---|---|---|---|---|---|---|---|
| **eps** | 0.05 | $78.41_{\pm 0.70}$ | $80.72_{\pm 1.23}$ | $75.04_{\pm 0.92}$ | $75.92_{\pm 0.99}$ | $83.31_{\pm 1.52}$ | $84.13_{\pm 1.41}$ |
| | 0.1 | $79.11_{\pm 0.78}$ | $82.05_{\pm 1.05}$ | $78.66_{\pm 0.96}$ | $79.51_{\pm 0.97}$ | $87.28_{\pm 1.61}$ | $87.62_{\pm 1.45}$ |
| | 0.2 | $79.34_{\pm 1.17}$ | $83.27_{\pm 1.60}$ | $75.59_{\pm 1.39}$ | $76.56_{\pm 1.50}$ | $85.48_{\pm 2.59}$ | $85.35_{\pm 2.94}$ |
| | 0.3 | $78.48_{\pm 0.61}$ | $82.65_{\pm 0.88}$ | $75.31_{\pm 0.69}$ | $76.28_{\pm 0.72}$ | $86.25_{\pm 1.31}$ | $86.97_{\pm 1.11}$ |
| **rho** | 0.55 | $77.01_{\pm 0.72}$ | $81.63_{\pm 1.15}$ | $72.72_{\pm 1.05}$ | $73.42_{\pm 1.08}$ | $82.47_{\pm 1.38}$ | $83.43_{\pm 1.29}$ |
| | 0.65 | $79.34_{\pm 1.17}$ | $83.27_{\pm 1.60}$ | $75.59_{\pm 1.39}$ | $76.56_{\pm 1.50}$ | $85.48_{\pm 2.59}$ | $85.35_{\pm 2.94}$ |
| | 0.75 | $78.89_{\pm 1.12}$ | $80.10_{\pm 1.32}$ | $75.77_{\pm 1.05}$ | $76.72_{\pm 1.08}$ | $83.92_{\pm 1.40}$ | $84.74_{\pm 1.34}$ |
| | 0.85 | $78.41_{\pm 1.03}$ | $78.65_{\pm 1.24}$ | $76.15_{\pm 0.87}$ | $76.81_{\pm 0.95}$ | $80.10_{\pm 1.10}$ | $80.25_{\pm 1.15}$ |
| **stddev** | 1.0 | $78.97_{\pm 0.71}$ | $81.65_{\pm 1.03}$ | $75.54_{\pm 0.70}$ | $76.47_{\pm 0.89}$ | $82.17_{\pm 0.98}$ | $82.37_{\pm 1.08}$ |
| | 2.0 | $79.34_{\pm 1.17}$ | $83.27_{\pm 1.60}$ | $75.59_{\pm 1.39}$ | $76.56_{\pm 1.50}$ | $85.48_{\pm 2.59}$ | $85.35_{\pm 2.94}$ |
| | 3.0 | $78.55_{\pm 0.63}$ | $79.82_{\pm 1.18}$ | $75.66_{\pm 0.91}$ | $76.48_{\pm 0.94}$ | $81.92_{\pm 1.12}$ | $81.33_{\pm 1.21}$ |
| | 4.0 | $78.83_{\pm 0.89}$ | $80.55_{\pm 1.02}$ | $75.76_{\pm 0.90}$ | $76.64_{\pm 1.03}$ | $81.94_{\pm 1.21}$ | $82.18_{\pm 1.36}$ |
| **Baseline** | Medformer | $76.99_{\pm 2.72}$ | $77.58_{\pm 4.09}$ | $74.82_{\pm 1.83}$ | $75.37_{\pm 2.22}$ | $82.93_{\pm 2.31}$ | $83.70_{\pm 2.08}$ |

Table 7. We investigated: *eps* (maximum perturbation magnitude), *rho* (adaptive step size parameter), and *stddev* (standard deviation of initial random noise). The highlighted rows indicate the optimal parameters used in our main experiments.

## D.2 STABILITY TO RANDOM INITIALIZATION

Figure 5 presents heatmaps of the standard deviation of performance metrics across five random seeds. Lower values indicate greater stability to initialization variance. SM demonstrates superior stability (lowest standard deviation) on the ADFTD and PTB datasets and consistently ranks among the top three most stable models on APAVA and TDBrain, all while achieving higher mean performance compared to other stable models like iTransformer.

## D.3 ROBUSTNESS TO GAUSSIAN NOISE

Tables 8 to 10 and Figures 6 to 8 detail the performance of SM and Medformer under increasing levels of Gaussian noise. While both models' performance degrades, SM consistently maintains higher accuracy and precision, especially at higher noise amplitudes ($\sigma > 1.0$). This demonstrates SM's superior ability to preserve predictive reliability in challenging noisy conditions, substantiating its enhanced robustness.

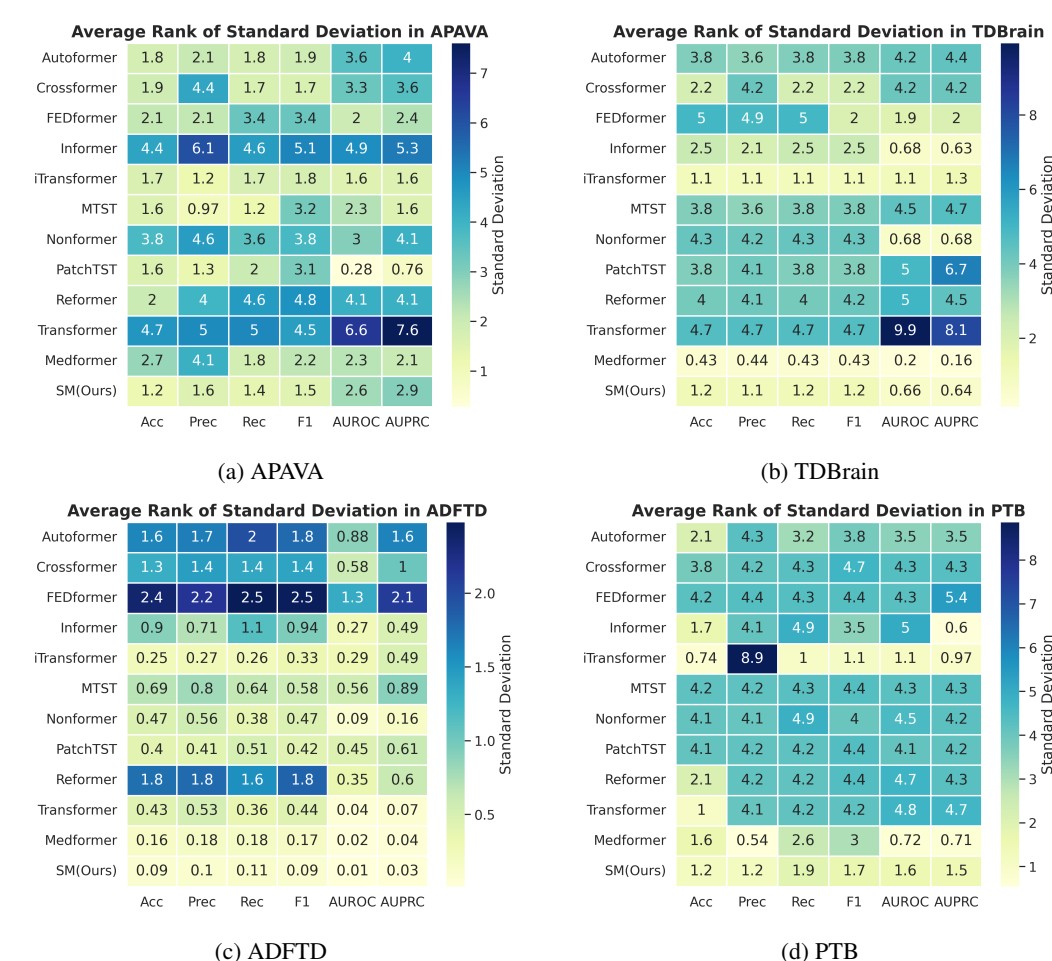

Figure 5: Average rank of standard deviation of performance across different random seeds for all models on the four datasets.

Table 8: Performance on ADFTD under varying Gaussian noise amplitudes.

| Model | Metric | Noise Amplitude ($\sigma$) | | | | | | | | |
|---|---|---|---|---|---|---|---|---|---|---|
| | | 0.1 | 0.5 | 1.0 | 1.5 | 2.0 | 2.5 | 3.0 | 3.5 | 4.0 |
| **SM** | Accuracy | 96.61 | 68.27 | 46.47 | 36.54 | 31.90 | 31.15 | 30.94 | 30.92 | 30.91 |
| | Precision | 96.47 | 68.07 | 51.08 | 45.29 | 40.85 | 37.44 | 37.23 | 36.39 | 35.44 |
| **Medformer** | Accuracy | 96.41 | 68.24 | 46.69 | 35.83 | 31.54 | 30.32 | 29.57 | 29.37 | 29.18 |
| | Precision | 96.19 | 67.46 | 51.36 | 44.30 | 39.90 | 37.35 | 36.49 | 34.93 | 34.92 |

## D.4 COMPUTATIONAL COST ANALYSIS

The enhanced robustness of SM comes with an increased computational cost during training due to the multifaceted stability optimization, particularly the adversarial sample generation. To quantify this, we compared the training time of our full SM model against the baseline Medformer.

On average, one epoch of training for SM required **approximately 1.8 times** the wall-clock time compared to the baseline. This overhead is primarily attributed to the forward and backward passes required by AutoPGD to craft adversarial examples. We argue that this is a justifiable trade-off for the substantial gains in model stability and reliability, which are paramount in safety-critical medical applications.

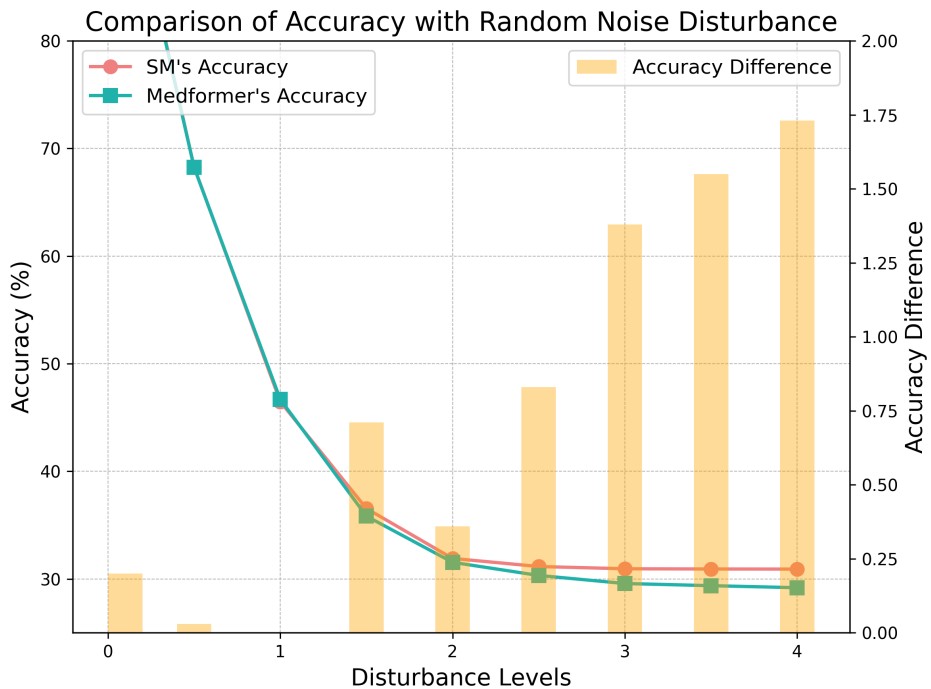

Figure 6: Comparison of accuracy and precision under Gaussian noise on the ADFTD dataset.

Table 9: Performance on PTB under varying Gaussian noise amplitudes.

| Model | Metric | Noise Amplitude ($\sigma$) | | | | | | | | |
|---|---|---|---|---|---|---|---|---|---|---|
| | | 0.1 | 0.5 | 1.0 | 1.5 | 2.0 | 2.5 | 3.0 | 3.5 | 4.0 |
| **SM** | Accuracy | 81.17 | 80.26 | 77.80 | 74.73 | 71.64 | 69.66 | 68.28 | 67.61 | 67.45 |
| | Precision | 85.88 | 85.76 | 85.35 | 84.70 | 82.98 | 81.89 | 79.65 | 78.25 | 79.00 |
| **Medformer** | Accuracy | 79.76 | 78.50 | 75.76 | 73.01 | 70.73 | 69.26 | 68.43 | 67.96 | 67.65 |
| | Precision | 87.14 | 86.65 | 85.49 | 83.80 | 82.37 | 79.86 | 76.43 | 71.39 | 67.65 |

Importantly, the architectural modifications in the SHSM module are lightweight. Therefore, the **inference time of SM is nearly identical** to that of Medformer, as the adversarial training components are not used during evaluation. This ensures that our model can be deployed without introducing significant latency.

## E   THE USE OF LARGE LANGUAGE MODELS(LLMS)

We acknowledge the use of large language models (LLMs) as auxiliary tools in the preparation and refinement of this manuscript. Their role was limited to grammar verification, stylistic improvement of expressions, and conversion of mathematical formulas.

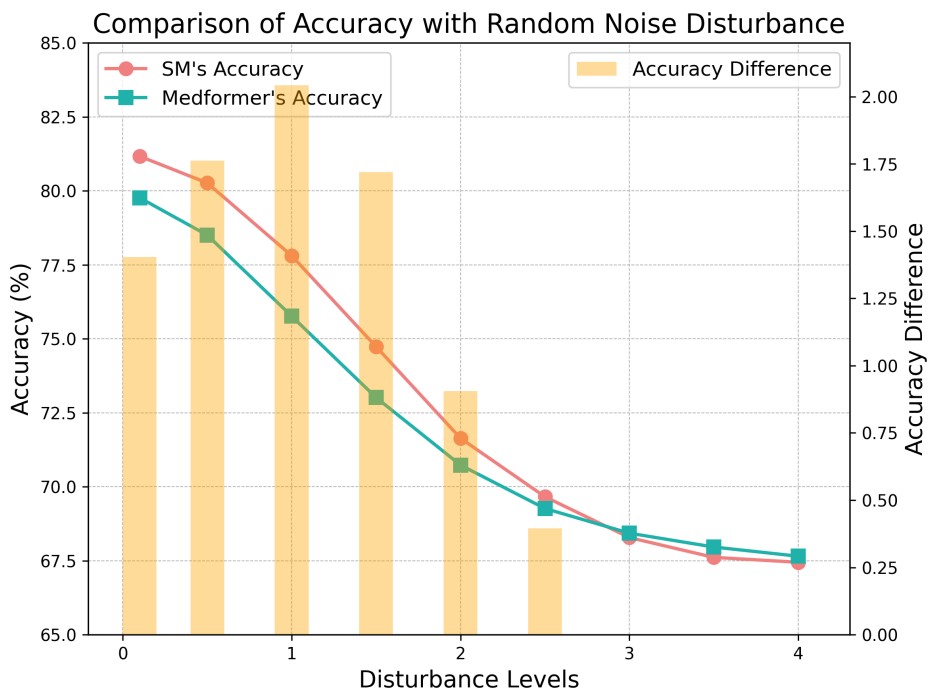

Figure 7: Comparison of accuracy and precision under Gaussian noise on the PTB dataset.

Table 10: Performance on TDBrain under varying Gaussian noise amplitudes.

| Model | Metric | Noise Amplitude ($\sigma$) | | | | | | | | |
|---|---|---|---|---|---|---|---|---|---|---|
| | | 0.1 | 0.5 | 1.0 | 1.5 | 2.0 | 2.5 | 3.0 | 3.5 | 4.0 |
| **SM** | Accuracy | 89.62 | 89.08 | 86.38 | 82.08 | 77.75 | 71.11 | 66.81 | 62.94 | 60.50 |
| | Precision | 89.72 | 89.18 | 86.47 | 82.16 | 77.79 | 71.17 | 66.95 | 63.10 | 60.69 |
| **Medformer** | Accuracy | 88.04 | 87.15 | 85.12 | 81.02 | 76.85 | 69.29 | 65.46 | 61.10 | 58.56 |
| | Precision | 88.16 | 87.28 | 85.36 | 81.33 | 77.25 | 69.91 | 66.20 | 61.75 | 59.37 |

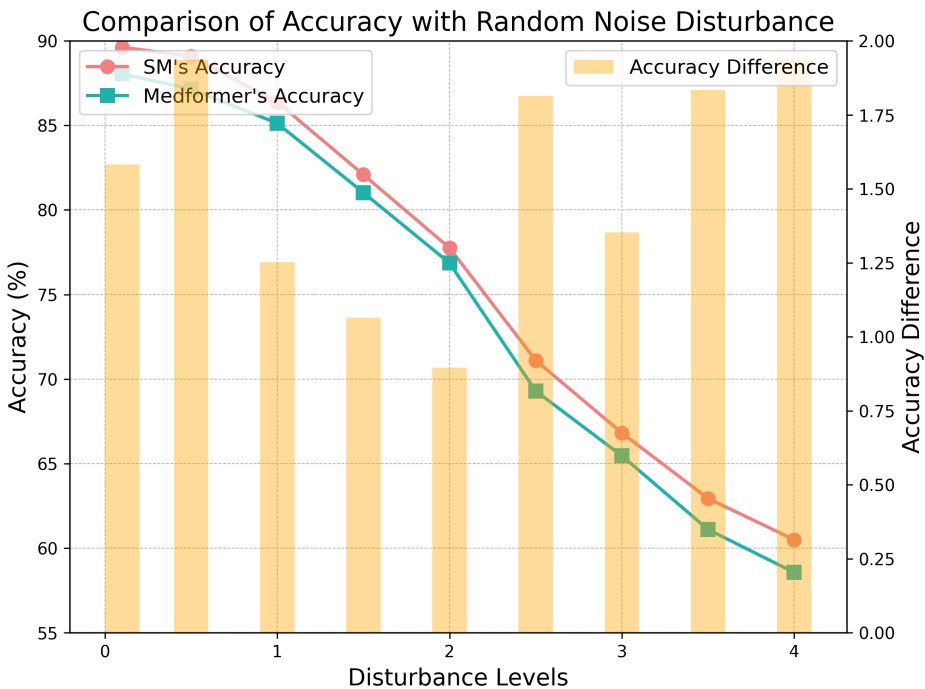

Figure 8: Comparison of accuracy and precision under Gaussian noise on the TDBrain dataset.

