# OpenReview forum: "SM: Bridging the Robustness Gap in Clinical Time Series Analysis via Hierarchical Stability Optimization"
_ICLR.cc/2026/Conference — ICLR 2026 Conference Withdrawn Submission_

### Official Review · Reviewer_p5N7 · 2025-10-21

**Soundness:** 1
**Presentation:** 3
**Contribution:** 3
**Rating:** 4
**Confidence:** 4

**Summary:**

In their submission, the authors address robustness issues in medical time series classification models, which is a relevant and timely challenge in medical time series analysis. Building on (most likely own) prior work in the form of MedFormer, a transformer-based medical time series classification model, they propose both architectural improvements (e.g. modifications of the attention mechanism) as well as improvements in the training procedure (e.g. adversarial training and consistency losses as well as training-teacher methods) that are supposed to enhance the robustness of the model. The authors compare their proposed model on 3 EEG and 1 ECG dataset against 11 transformer-based baselines, where the model achieves the best performance across all of them. The authors provide ablation studies to investigate the effect of different components of the proposed approach and also investigate the stability of the approach in different scenarios such as scaling the size of the training set.

**Strengths:**

* The aspect of robustness of medical time series classifiers is a relevant and important topic for the clinical application of such systems (see also the corresponding weaknesses)
* The authors benchmark their model across multiple datasets covering two modalities and compare to a large number of baselines (albeit all of them transformer-based)
* The authors carry out ablation studies to demonstrate the importance of different solution components.

**Weaknesses:**

* Concerning the robustness motivation, the description stays fairly unspecific as to what kind of robustness represents the most significant challenge (measurement noise, distribution shift through different devices, movement artifacts,...) e.g. "even though the description stays fairly unspecific as to what kind of robustness represents the most significant challenge" The problem that arises from such an unspecific problem definition is that it is very difficult to test if the proposed model actually exhibits improved robustness, in-distribution tests based on predefined datasets are not sufficient for that. The authors fail to deliver clear evidence that actual state-of-the-art models in the respective domains actually have robustness issues.
* They highlight Medformer as state-of-the-art models neglecting comparison to state-of-the-art models in the respective fields, which are in most cases not transformer-based but rather mostly CNNs or potentially state-space-models. This point in the introduction is also reflected in the experimental setup, where the authors only compare to transformer-based baselines and not on widely used benchmark datasets, where comparison to literature results is straightforward.
* The lack of integration of hierarchical information is motivated by comparing QRS complexes as example for high-frequency components to delta waves as examples for low-frequency components, which mixes ECG and EEG concepts. This challenge already exists within a single modality, e.g. ECG analysis, considering long-range correlations across beats and high-frequency QRS complexes. Also, the authors mention "silent failures" that could occur during training, but the provided references do not support this argument.
* There are further unsubstantiated claims like "existing architectures tend to flatten multi-scale temporal interactions, rendering biomarker representations vulnerable to localized noise", which would, if evidence was provided, align nicely with the proposed solution.
* Subject-specific overfitting: Commonly used benchmark datasets in both considered signal domains typically avoid data leakage by avoiding shared subjects across train and test.
* Motivation of architectural design: "channels carrying critical diagnostic information (e.g., a QRS complex in ECG) often
exhibit higher signal energy" does not seem to be a substantiated claim.
* All four datasets are comparably small, even though much larger, publicly available, established benchmark datasets exist for ECG (PTB-XL, Chapman, CODE-15,...) and EEG (e.g. TUH-EEG). This raises several issues: (1) the observed results might only apply to small-scale datasets (2) Due to the inability to compare to established benchmarking results, makes it difficult to judge whether any of the presented results actually achieve state-of-the-art performance.
* In the results tables, the authors indicate model performances with confidence intervals. In case of overlapping confidence intervals, there is no way to assess the statistical significance of their finding. It would be insightful to assess this through pairwise statistical tests or via bootstrapping of the performance difference. This might also impact the ablation study, where many architectural changes could actually turn out to lead to non-significant effect.

The most significant weaknesses are (1) the rather unspecific robustness definition and the lack of an appropriate evaluation methodology (2) the choice of benchmark datasets (3) the lack of a statistical significance analysis.

**Questions:**

* Consistency loss: We did the authors use MSE instead of a KL-divergence, which is typically used in such contexts? [1]
* Did the authors perform experiments on larger, widely used datasets e.g. CODE-15, PTB-XL, TUH-EEG etc?

[1] Zheng, S., Song, Y., Leung, T., & Goodfellow, I. (2016). Improving the robustness of deep neural networks via stability training. In Proceedings of the ieee conference on computer vision and pattern recognition (pp. 4480-4488).

---

### Official Review · Reviewer_5Tqs · 2025-10-30

**Soundness:** 2
**Presentation:** 3
**Contribution:** 2
**Rating:** 4
**Confidence:** 5

**Summary:**

This manuscript proposes SM (Stability Medical time series classifier), a framework designed to enhance the robustness of deep learning models for medical time series analysis, particularly focusing on electrocardiography (ECG) and electroencephalography (EEG) data. The core contributions include a novel Stability-aware Hierarchical Spatial Modulation (SHSM) module that mimics clinical reasoning by selectively attending to biomarkers while preserving global waveform morphology, and a multifaceted stability optimization strategy that combines adversarial training, output consistency, and knowledge distillation. The framework is evaluated on four medical datasets (APAVA, TDBrain, ADFTD, and PTB) and demonstrates state-of-the-art performance compared to 11 baseline models, with emphasis on bridging the gap between laboratory accuracy and clinical reliability.

**Strengths:**

1. The work creatively integrates physiological insight with modern deep learning techniques. The SHSM module is a novel architectural innovation that emulates clinical diagnostic focus, and the co-design of architecture and training objectives for stability is a distinctive approach not commonly seen in prior works.
2. The experimental evaluation is comprehensive, including comparisons with 11 strong baselines, ablation studies, robustness evaluations under noise, and few-shot learning scenarios. The use of multiple datasets enhances the generalizability of the findings.

**Weaknesses:**

1. The clinical rationale for the core SHSM module is a significant concern. Its energy-based channel selection operates on the assumption that higher signal energy correlates with greater diagnostic importance, which is an oversimplification that may not hold in practice. For example, in a standard 12-lead ECG, voltage is largely determined by lead placement and anatomy, not diagnostic value. More critically, many crucial biomarkers are inherently low-energy, such as subtle fibrillatory waves in Atrial Fibrillation or minor ST-segment changes in ischemia. Moreover, "intuition is that channels carrying critical diagnostic information (e.g., a QRS complex in ECG) often exhibit higher signal energy" is misleading, as the QRS complex is present in all leads.
2. The novelty of the methodology appears to lie more in the systematic integration of existing techniques than in fundamental innovation. The components of the stability optimization strategy—adversarial training with AutoPGD, output consistency loss, and knowledge distillation—are well-established in the broader robustness literature. While their careful co-design and application to medical time series is a contribution, the paper would benefit from a more precise framing that highlights this integrative synthesis as the primary advancement.
3. The ECG dataset (PTB) used for evaluation is relatively small and not so up-to-date compared to larger, more diverse datasets like PhysioNet/CinC Challenge datasets, e.g. 2020/2021 and 2025 this year (containing the CODE-15% dataset which is one of the largest ECG datasets available). Evaluating on such datasets would strengthen the claims of generalizability and clinical relevance.

**Questions:**

See the weaknesses section.

---

### Official Review · Reviewer_2Zdw · 2025-10-31

**Soundness:** 2
**Presentation:** 2
**Contribution:** 2
**Rating:** 4
**Confidence:** 5

**Summary:**

This paper presents Stability-aware Hierarchical Spatial Modulation (SHSM) to address the challenge of classifying noisy real-world medical time series (MedTS) trained on clean data. The method uses a teacher-student architecture in which the teacher is trained on clean data and the student on data with injected noise. Four types of losses are used to ensure the performance of clean data and robustness against adversarial noise. Experiment performed on 4 datasets against 11 baselines.

**Strengths:**

The motivation is good, given the inconsistency between lab-clean data and real-world noisy data. The teacher-student structure for learning by gradually adding noise to build a noise-robust model is interesting.

**Weaknesses:**

1) The motivation for building a noisy-robust model for real-world data is good; however, simply adding Gaussian noise does not sound very practical. For example, in EEG, artifact noise, such as eye blinks, muscle movement, and heartbeat, is not well approximated by a Gaussian distribution.
2) The energy-based channel selection is problematic. The energy of a channel does not equal the informative channels in the EEG. Some channels are energicatic because the devices are easier to collect, such as Fp1 and Fp2. But in the meantime, they are bothered by the eye-click artifacts issue. Besides, the low energy of some channels might be vital for EEG classification. Simply removing the low-energy channel is arbitrary and does not match the neuroscience knowledge.
3) Although there is a mismatch between clean data and real-world data in medical time series, we could still alleviate the gap by using the same device for collecting and following the same pipeline to clean it. The model should still work even without manual cleaning during training.
4) The experiment results are not SOTA. Many performances are lower than the previously reported results. Especially for ADFTD, a subject-dependent setup should not be used formally, as shortcut learning in EEG-based brain disease detection leads to substantial performance inflation.
5) For future improvement, authors may try using the designed structure for large self-supervised pre-training, such as the EEG/ECG foundation model. As the 4 datasets are relatively small, it might not be a good idea to combine such a complex structure and train them fully supervised.

**Questions:**

See weakness.

---

### Official Review · Reviewer_6kTh · 2025-11-06

**Soundness:** 3
**Presentation:** 3
**Contribution:** 2
**Rating:** 4
**Confidence:** 4

**Summary:**

The authors try to solve a critical problem: AI models for medical diagnosis (EEG, ECG) work well in labs but fail in real clinics due to noise and device variability. They propose SM, a framework that makes models simultaneously accurate and robust through two innovations. Architecturally, the SHSM module mimics clinical reasoning by selectively focusing on diagnostically important signal channels via sparse attention while preserving overall waveform morphology through lightweight convolutions. For training, a multifaceted optimization strategy combines four complementary losses—standard classification, adversarial robustness, output consistency, and knowledge distillation—to enforce both accuracy and stability.

**Strengths:**

1. The robustness gap between lab and clinic is genuine and important for medical AI deployment.
2. Comprehensive experiments clearly demonstrate improved robustness while maintaining accuracy. The ablation studies convincingly show each component contributes value.

**Weaknesses:**

1. Limited novelty. Individual components (adversarial training, knowledge distillation, sparse attention) are well-known. The contribution is their combination, not fundamental innovation.

2. The paper evaluates each dataset in isolation. This tests whether models generalize to new subjects within the same acquisition environment, not whether they generalize across different devices, protocols, or institutions. Cross-dataset evaluation (e.g., train on APAVA, test on TDBrain) would be necessary to validate claims about device/institutional robustness. Without this, the paper demonstrates improved stability to noise within a controlled environment, but not robustness to the systematic differences between real-world clinical deployments.

3. Unrealistic noise testing. Gaussian noise and AutoPGD perturbations don't capture real clinical artifacts like motion artifacts or electrode impedance changes. This undermines claims about clinical deployability.

**Questions:**

1. Would this work with real clinical noise? The paper needs validation with actual recorded artifacts, not synthetic perturbations.
2. What is the model actually learning? No visualization of which patterns SHSM attends to - is it really finding clinically meaningful features?

---

### Note · Authors · 2026-01-06

I have read and agree with the venue's withdrawal policy on behalf of myself and my co-authors.